# Factors associated with mortality in patients hospitalized for COVID-19 admitted to a tertiary hospital in Lambayeque, Peru, during the first wave of the pandemic

Edwin Aguirre-Milachay[1,2], Darwin A. León-Figueroa[2,3], Marisella Chumán-Sánchez[2,4], Luccio Romani[2,3], Fernando M. Runzer-Colmenares[5]*

1 Servicio de Geriatría, Departamento de Medicina, Hospital Nacional Almanzor Aguinaga Asenjo, Chiclayo, Peru, 2 Facultad de Medicina Humana, Universidad de San Martín de Porres, Chiclayo, Peru, 3 Emerge, Unidad de Investigación en Enfermedades Emergentes y Cambio Climático, Facultad de Salud Pública y Administración, Universidad Peruana Cayetano Heredia, Lima, Peru, 4 Sociedad Científica de Estudiantes de Medicina Veritas (SCIEMVE), Chiclayo, Perú, 5 CHANGE Research Working Group, Carrera de Medicina Humana, Universidad Científica del Sur, Lima, Peru

* frunzer@cientificadelsur.edu.pe

## Abstract

### Introduction

COVID-19 caused by Severe Acute Respiratory Syndrome Coronavirus 2 (SARS-CoV-2) has spread worldwide, becoming a long-term pandemic.

### Objectives

To analyze the factors associated with mortality in patients hospitalized for COVID-19 in a tertiary hospital in the Lambayeque region of Peru.

### Methods

A retrospective cohort study of patients with a diagnosis of COVID-19, hospitalized in a hospital in northern Peru, was conducted from March to September 2020.

### Results

Of the 297 patients studied, 69% were women, the mean age was 63.99 years (SD = ±15.33 years). Hypertension was the most frequent comorbidity (36.67%), followed by diabetes mellitus (24.67%) and obesity (8.33%). The probability of survival at 3 days of ICU stay was 65.3%, at 7 days 24.2%, and 0% on day 14. Risk factors associated with mortality in patients hospitalized for COVID-19 are age, male sex, tachypnea, low systolic blood pressure, low peripheral oxygen saturation, impaired renal function, elevated IL-6 and elevated D-dimer.

### Conclusions

Mortality in hospitalized patients with COVID-19 was 51.18 per 100 persons, Mortality was found to be associated with hypertension, type of infiltrating, and sepsis.

**Data Availability Statement:** The dataset is publicly available and was uploaded to the Figshare platform (https://doi.org/10.6084/m9.figshare.21664172.v1), the information was encrypted to protect the privacy of the research participants.

**Funding:** This research received no external funding.

**Competing interests:** The authors have declared that no competing interests exist.

# 1. Introduction

Coronavirus Disease 2019 (COVID-19) caused by Severe Acute Respiratory Syndrome Coronavirus 2 (SARS-CoV-2) has spread globally, becoming a long-lasting pandemic [1]. By December 1, 2022, the World Health Organization (WHO) has reported more than six million deaths due to COVID-19 [2], in Peru, more than 217,370 deaths due to the disease have been reported by 2022, with an estimated case fatality rate of 5.14% [3].

Risk factors associated with mortality in patients hospitalized for COVID-19 are age, male sex, tachypnea, low systolic blood pressure, low peripheral oxygen saturation, impaired renal function, elevated IL-6, elevated D-dimer, and elevated troponin [4, 5]. In addition, the presence of comorbidities such as hypertension, coronary heart disease, and diabetes was associated with a significantly increased risk of death among patients with COVID-19 [4, 6].

About 5–32% of patients hospitalized for COVID-19 require admission to the intensive care unit (ICU), mainly for acute hypoxemic respiratory failure [7–10]. Acute respiratory distress syndrome (ARDS) is diagnosed in 40–96% of ICU patients and 30–88% of them need invasive mechanical ventilation (IVM) [9–15].

Reported ICU mortality ranges from 16 to 78% depending on patient and health system characteristics and the percentage of patients still in the ICU at the end of follow-up [9, 11–13, 15, 16]. In particular, ICU mortality in patients receiving IVM is consistently higher, reaching values of up to 88% [17].

The aim of this study was to analyze the factors associated with mortality in patients hospitalized for COVID-19 in a tertiary hospital in the Lambayeque region. For this purpose, we describe the clinical, epidemiological, laboratory, and imaging characteristics, pharmacological treatments, and ventilatory support in patients hospitalized for COVID-19, which allows us to establish a prognostic model based on the diversity of parameters studied that helps us to strengthen the main variables except for immunization, which influenced the main adverse outcome in a Peruvian population during the first wave of COVID-19, relevant information for future outbreaks in non-immunized population or with incomplete vaccination schedules.

# 2. Materials and methods

## 2.1. Study design and study population

A retrospective cohort study was conducted during the first wave of the COVID-19 pandemic in Peru at the Hospital Nacional Almanzor Aguinaga Asenjo (HNAAA). The HNAAA is a hospital of the third level of care of the Peruvian Social Security (EsSalud), located in the city of Chiclayo, Peru.

The study period comprised the months of March to September 2020, corresponding to the first wave of deaths due to COVID-19 in Peru, which included the first peak of mortality due to the disease.

The sample was obtained based on the study by Hueda-Zavaleta et al [18], using the EPI-DAT 4 statistical package. 2, determined based on oxygen saturation ($< 90$%) with a proportion of exposed 30.3% and a proportion not exposed of 13.773%, with a non-exposed/exposed index of 1, a confidence level of 95%, a power of 80%, a significance level of 5%. The sample obtained was 196, adding a 20% loss to the data, we obtain 235.

The study included all patients diagnosed with COVID-19, by serological rapid test (RP) (IgM and/or IgG) or molecular test by reverse transcriptase polymerase chain reaction (RT-PCR) analysis, hospitalized at HNAAA during the study period.

## 2.2. Data collection

Data were collected by reviewing the patient's electronic medical records. The variables of interest were recorded in a Microsoft Excel ® spreadsheet independently by two authors. The outcome was hospital discharge or death of the patients.

In addition, age, sex, time of illness, COVID-19 diagnostic method, laboratory characteristics at hospital admission, imaging data, mortality, and ICU admission criteria defined by the presence of 3 or more criteria were considered: Moderate-severe dyspnea with signs of work of breathing and use of accessory musculature or paradoxical abdominal movement, FR>30, o2 saturation less than 92 with use of high flow oxygen therapy, PAFIO2 less than 150, Lactate greater than 2mm/l or capillary refill greater than 2 sec without response to fluid therapy, MAP less than 65mmHg without response to fluid therapy, Organ dysfunction: mental confusion, renal, cardiological and/or hepatic abnormality, Chest CT scan with greater than 50% compromise.

In addition, days from diagnosis to ICU admission, comorbidities, days of ventilatory support, treatment before hospitalization, in-hospital treatment, complications such as Mild-Moderate Respiratory Distress Syndrome, Severe Respiratory Distress Syndrome, Sepsis/Septic Shock, which were determined by an intensivist physician and obtained through medical records during follow-up, and days of ICU stay.

## 2.3. Statistical analysis

For the descriptive analysis, we used absolute and relative frequencies, measures of central tendency and dispersion, previously evaluating normality with histograms and Shapiro-Wilk. The bivariate analysis was performed using the Chi-square test or Fisher's exact test for categorical variables, and Mann-Whitney U for ordinal variables based on a $p<0.05$ (95% confidence level).

To assess the association between independent variables and mortality, prevalence ratios (PR) and their respective 95% confidence intervals (95%CI) were calculated using Poisson regressions with robust variance. The Kaplan-Meier test was used to calculate mortality by estimating the probability of survival of patients with COVID-19. The statistical package STATA version 17 (StataCorp, College Station, TX, USA) was used for data processing.

## 2.4. Ethical information

The protocol of the present study was approved by the Research Ethics Committee of the Hospital Nacional Almanzor Aguinaga Asenjo, Lambayeque-Perú. The registration number is CIE-RPL:071-DIC-2021. This protocol complied with the norms contained in the Helsinki Declaration. The data obtained were used only for research purposes, preserving the anonymity of the participants, and eliminating data that could be identified. The database of the present study will be deleted two years after its publication.

## 3. Results

We reviewed the data of 300 patients hospitalized at the Hospital Almanzor Aguinaga of the Red Lambayeque del Seguro Social of Peru who was followed during March and April, August, and September 2020, 3 medical records were excluded for not having the outcome. It was decided to work with 297 patients, being higher than the calculated sample size, to improve the results found.

## 3.1. General characteristics of the population and bivariate analysis

Of the 297 patients studied, 69% were women, the mean age was 63.99 years (SD = ±15.33 years). Hypertension was the most frequent comorbidity (36.67%), followed by diabetes mellitus (24.67%) and obesity (8.33%). The median time of illness was 7 days (RIC: 4–11 days), 80.33% presented fever, and 83% had blood group type O (Table 1).

Regarding pulmonary involvement, the mean respiratory rate was 22.5 breaths per minute. The median oxygen saturation on admission was 88% (RIC: 76–93%) in patients without oxygen therapy and 93% (RIC: 89–96%) in patients with oxygen therapy. 65.22% had a ground-glass infiltrate, 74.45% had a bilateral distribution and 51.49% had lung involvement greater than or equal to 50% (Table 1).

A total of 152 (51.18%) died during their hospital stay, representing a rate of 51.18. Statistical significance ($p < 0.05$) was found concerning mortality with epidemiological variables such as age, oxygen saturation on admission with oxygen therapy, respiratory frequency on admission, comorbidities such as hypertension, diabetes mellitus, and symptoms reported on admissions such as cough and dyspnea. Other general characteristics are shown in Table 1.

The diagnosis of Covid-19 was mostly made with the rapid test in 188 patients, PCR-RT in 62, and antigenic test in 1 patient. The remaining patients were probable diagnoses based on clinical and imaging findings. Within the laboratory results at admission, it was found that within the coagulation profile the median fibrinogen level was 929 (RIC:504.7–1505 md/dl), and the median prothrombin time was 11 (RIC: 10.40–12); likewise, it was found that the median C-reactive protein levels were 2.9 mg/dl (RIC: 0.9–11.01), ferritin was 929.9 ng/ml (RIC: 504. 7–1505), D-dimer was 1.77 mcg/ml (RIC: 0.84–4.53). Regarding arterial gas analysis, the mean PH at admission was 7.39 (SD: ±0.098), median O2 partial pressure was 68.5 (54.5–89), and CO2 was 36(31–42).

Among the laboratory results in hospitalization, it was found that within the coagulation profile the median fibrinogen level was 520 (RIC: 430–650 md/dl), and the median prothrombin time was 11 (RIC: 10.20–11.90); also the median C-reactive protein levels were found to be 8.90 mg/dl (RIC: 4.00–20.00), ferritin was 1096 ng/ml (RIC: 604.30–1745), D-dimer was 1.71 mcg/ml (0.86–3.38). The other laboratory test results are listed in Table 2.

In the treatment received, Tocilizumab was only used in one patient, which resulted in mortality. Among the drugs used before hospitalization, we found that ivermectin was used in 23.33% of the patients, azithromycin in 26.67%, and corticoids in 23.33%. In hospitalization we found that ivermectin was used in 25.67% of the patients, azithromycin in 67.33%, standard doses of corticoids were used in 65.33% of the patients, and 53.33% of the patients received therapeutic doses of anticoagulants.

Statistical significance was found concerning outcomes with the use of corticosteroids and anticoagulants before hospitalization. Concerning the drugs used in hospitalization, statistical significance was found in the use of antibiotics, corticoids, and anticoagulants in therapeutic doses. The other results are shown in Table 3.

## 3.2. Characteristics of the treatment received and complications

The most frequent complications reported were: Acute Respiratory Distress Syndrome (ARDS) according to SAFI less than 150 in 151 (50.8%) patients, in those with hospital mortality it reached 121 (80.1%), presenting statistical significance. Mild-moderate ARDS was present in 120 (40.4%) hospitalized patients, 99 (82.5%) without hospital mortality, the result was also statistically significant. Likewise, we found statistical significance with the presence of sepsis, which was present in 152 (51.1%) hospitalized patients (Table 3).

Oxygen therapy times were also evaluated. The average time of binasal cannula use was 1.86±3.29 days with an inverse correlation with time to mortality (spearman -0.6003 $p < 0.000$),

**Table 1. General characteristics of the study population.**

| Variables | Total | Death | | p-value[¥] |
|---|---|---|---|---|
| | n (%) | Yes | No | |
| Age (years) | 63.99±15.33* | 69.9±14.03* | 57.4 ±13.9* | <0.001 |
| Sex | | | | |
| Female | 207 (69) | 103 (50.49) | 101 (49.51) | 0.725 |
| Male | 93 (31) | 49 (52.79) | 44 (47.31) | |
| Time of illness (days) | 7 (4–11)** | 7 (4.5–12)** | 7 (4–11)** | 0.580 |
| Intake O2 saturation (%) | | | | |
| Without oxygen therapy | 88 (76–93)** | 77.5 (70–87)** | 92 (89–95)** | <0.001 |
| With oxygen therapy | 93 (89–96)** | 92 (85.5–95)** | 95 (92–96)** | |
| Respiratory frequency (x min) | 22.5±4.17* | 24.37±4.15* | 22.13±3.94* | <0.001 |
| **Comorbidities** | | | | |
| Arterial hypertension | 110 (36,67) | 77 (70,64) | 32 (29,36) | <0,001 |
| Diabetes mellitus | 74 (24.67) | 47 (64.40) | 26 (35.60) | <0.001 |
| Cardiovascular disease | 13 (4.33) | 10 (76.92) | 3 (23.08) | 0.086 |
| Obesity | 25 (8.33) | 13 (54.17) | 11 (45.83) | 0.760 |
| COPD | 9 (3.00) | 4 (50.00) | 4 (50.00) | 0.946 |
| Asthma | 12 (4.00) | 3 (25.00) | 9 (75.00) | 0.064 |
| Cancer | 14 (4.68) | 4 (30.77) | 9 (69.23) | 0.161 |
| **Symptoms** | | | | |
| Fever | 173 (57.67) | 92 (53.49) | 80 (46.51) | 0.350 |
| Cough | 171 (57.00) | 90 (54.25) | 79 (46.75) | 0.029 |
| Dyspnea | 241 (80.33) | 131 (54.58) | 109 (45.42) | 0.016 |
| Diarrhea | 26 (8.67) | 8 (30.77) | 18 (69.23) | 0.411 |
| **Blood Group** | | | | |
| Blood group 0 | 83 (76.85) | 43 (52.44) | 39 (47.56) | 0.964 |
| Blood group A | 18 (16.67) | 10 (58.82) | 7 (41.18) | |
| Blood group B | 6 (5.56) | 3 (50) | 3 (50) | |
| Blood group AB | 1 (0.93) | 1 (100) | - | |
| **Type of Infiltrator** | | | | |
| Tarnished glass | 105 (65.22) | 37 (35.6) | 67 (64.4) | **0.027** |
| Cobblestone | 27 (16.77) | 16 (59.3) | 11 (40.70) | |
| Consolidation | 6 (3.73) | - | 6 (100) | |
| Mixed | 23 (14.29) | 8 (34.8) | 15 (65.22) | |
| **Distribution** | | | | |
| No Infiltrate | 45 (22.73) | 21 (46.67) | 24 (53.33) | 0.479 |
| Unilateral | 5 (2.53) | 1 (20.00) | 4 (80.00) | |
| Bilateral | 148 (74.75) | 60 (40.82) | 87 (59.18) | |
| **Pulmonary involvement** | | | | |
| >50% | 69 (51.49) | 40 (57.97) | 29 (42.03) | **<0.001** |
| <50% | 65 (48.51) | 11 (17.19) | 53 (82.81) | |

* Mean and standard deviation

** Median and interquartile ranges (25–75%)

[¥] P-values calculated using the Chi$^2$ test, Fisher's exact test, T-student, U Mann Whitney, as appropriate.

**Table 2. Laboratory tests of the studied population.**

| Variables | Total | Death | | p-value¥ |
|---|---|---|---|---|
| | n (%) | Yes | No | |
| **Upon entry** | | | | |
| Leukocytes (x1000) | 9.84 (8.02–14.19) ** | 16.93 (11.41–21.43)** | 8.46 (7.31–10.43) ** | <0.001 |
| MV/MIV lymphocytes (%) | 10.05 (4–20.10) ** | 3.55 (2–5) ** | 17.50 (11–24.70) ** | <0.001 |
| Platelets (x 1000) | 292 (196–387) ** | 260 (175–334)** | 313 (212–406) ** | 0.057 |
| PCR (mg/dl) | 2.9 (0.9–11.01) ** | 14.8 (6.70–23) ** | 1.10 (0.30–2.85) ** | <0.001 |
| Ferritin (ng/ml) | 929.9 (504.7–1505) ** | 1328 (780.1–2000) ** | 678.5 (373.35–1167,5) ** | 0.001 |
| Glucose (mg/dl) | 108 (86–146) ** | 129 (109–182) ** | 98 (82–117) ** | <0.001 |
| Creatinine (mg/dl) | 0.69 (0.57–0.96) ** | 0.76 (0.56–1.72) ** | 0.67 (0.57–0.81) ** | 0.171 |
| Urea (mg/dl) | 36 (23.20–60.10) ** | 59 (36.7–110) ** | 25.9 (21–40.40) ** | <0.001 |
| LDH (IU/l) | 289.90 (212.20–407.50)** | 458.80 (355.20–606.40) ** | 222 (189–258.50) ** | <0.001 |
| D-dimer (mcg/ml) | 1.77 (0.84–4.53) ** | 4.72 (2.13–5.00) ** | 1.06 (0.54–2.72) ** | <0.001 |
| Prothrombin time | 11 (10.40–12) ** | 11.72 (10.70–12.40) ** | 10.70 (10.10–11.50) ** | 0.002 |
| Fibrinogen | 451 (338.80–595.80)** | 523.50 (341.25–645.95)** | 426.85 (338.80–513.50)** | 0.017 |
| Ph at the entrance | 7.39(±0.098) * | 7.37 (±0.105) * | 7.43 (7,39–7.46) * | 0.012 |
| PO2 | 68.5 (54.5–89) ** | 62 (50–75) ** | 92 (68.5–123.5) ** | <0.001 |
| PCO2 | 36 (31–42) ** | 35 (31–44) ** | 37.5 (34–41) ** | 0.422 |
| **In Hospitalization** | | | | |
| Leukocytes (x 1000) | 10.86 (7.6–15.11) ** | 12.95 (9.77–16.73) ** | 9.22 (6.81–12.01) ** | <0.001 |
| Lymphocytes (%) | 8 (4–13.7) ** | 5 (3–8) ** | 12.9 (7–19) ** | <0.001 |
| Platelets (x 1000) | 261 (196–342) ** | 249 (163–303) ** | 286 (216–359) ** | 0.001 |
| PCR (mg/dl) | 8,90 (4–22,20)** | 16.3 (6.7–27.7)** | 5.2 (1.7–8.9)** | <0.001 |
| Ferritin (ng/ml) | 1096 (604.3–1745)** | 1295 (803.8–2000)** | 819 (498.2–1498)** | <0.001 |
| Glucose (mg/dl) | 116 (95–158)** | 127 (101–165)** | 111 (90–149)** | <0.001 |
| Creatinine (mg/dl) | 0.77 (0.63–1.12)** | 0.86 (0.67–1.39)** | 0.7 (0.58–0.88)** | 0.001 |
| Urea (mg/dl) | 43 (28.6–65.5)** | 52.6 (38.6–79)** | 35 (24.3–49)** | <0.001 |
| LDH (IU/l) | 328.75 (256.2–467.9)** | 441.7 (327.7–584.7)** | 267.9 (217–333.8)** | <0.001 |
| D-dimer (mcg/ml) | 1.71 (0.86–3.38)** | 2.32 (1.35–4.49)** | 1.36 (0.56–2.47)** | 0.001 |
| Prothrombin time | 11 (10.2–11.9)** | 11.5 (10.55–12)** | 10.9 (10–11.6)** | 0.007 |
| Fibrinogen | 520 (430–650)** | 520.5 (430–640)** | 519.9 (428.9–650)** | 0.936 |

* Mean and standard deviation

** Median and interquartile ranges (25–75%)

¥ P-values calculated using the T-student, U Mann Whitney, as appropriate.

and the average time of reservoir mask use was 3.44±4.26 days with a positive correlation with time to mortality (spearman 0.419 p<0.000). There was no correlation between the time of use of noninvasive ventilation and time to death (p = 0.31).

The average time of use of invasive mechanical ventilation was 0.30±1.51 days with a maximum value of 14 days, and a positive correlation was found with the time to death (spearman 0.228 p = 0.0001) (Fig 1).

Of the 297 hospitalized patients studied, 152 had mortality as the outcome and 145 were discharged or transferred to another service. Twenty-six patients were admitted to the Intensive Care Unit (ICU). The mean time to death was 2.66±4.2 days with a minimum of 0 and a maximum of 26 days and the average length of stay in the ICU was 0.49±1.86 days with a minimum of 0 and a maximum of 14 days.

**Table 3.** COVID-19 treatment used in the study population.

| Variables | Total | Death | | p-value¥ |
|---|---|---|---|---|
| | n (%) | Yes | No | |
| **Prior to hospitalization** | | | | |
| Ivermectin | 67 (22.33) | 37 (56.06) | 29 (43.94) | 0.368 |
| Azithromycin | 80 (26.67) | 41 (51.25) | 39 (48.75) | 0.988 |
| Hydroxychloroquine | 5 (1.67) | 3 (60.00) | 2 (40.00) | 0.691 |
| Corticosteroid | 70 (23.33) | 43 (62.32) | 26 (37.68) | 0.035 |
| Antibiotics | 79 (26.33) | 46 (58.97) | 32 (41.03) | 0.109 |
| Anticoagulants | 50 (16.67) | 32 (65.31) | 17 (34.69) | 0.030 |
| **In hospitalization** | | | | |
| Ivermectin | 77 (25.67) | 44 (57.89) | 32 (42.11) | 0.174 |
| Hydroxychloroquine | 70 (23.33) | 33 (47.14) | 37 (52.86) | 0.440 |
| Azithromycin | 202 (67.33) | 107 (52,97) | 95 (47.03) | 0.368 |
| Lopinavir/ritonavir | 53 (17.67) | 24 (45.28) | 29 (54.72) | 0.343 |
| Antibiotics | 278 (92.67) | 147 (52.88) | 131 (47.12) | 0.025 |
| Corticosteroids | | | | |
| Standard dosage | 196 (65.33) | 110 (56.7) | 84 (43.3) | 0.002 |
| Pulse dose | 42 (14.00) | 23 (54.8) | 19 (45.24) | |
| I do not use | 62 (20.67) | 19 (31.2) | 42 (68.9) | |
| **Anticoagulants** | | | | |
| Prophylactic Dose | 130 (43.33) | 82 (51.9) | 76 (48.1) | 0.791 |
| Therapeutic Dose | 160 (53.33) | 82 (63.6) | 47 (36.4) | <0.001 |

¥ P-values calculated using the Chi$^2$ test, Fisher's exact test, as appropriate.

The average time to discharge was 4.1±6.55 days and the average hospital stay was 6.66± 6.2 days with a minimum of 1 and a maximum of 48 days.

### 3.3. Survival analysis

The probability of survival at 3 days of hospitalization was 54.4%, at 7 days 23.5%, and 0% at day 26. In patients who were admitted to the ICU, the probability of survival on day 2 was 82.4% compared with those who were not admitted to the ICU, which was 65.9%; on day 7 it was 35.3% vs. 21.9%. This has minimal statistical significance (log-rank p = 0.053) (Fig 2).

In patients with Severe Respiratory Distress Syndrome determined by SAFI<150, the probability of survival on day 2 was 67.7% compared to those not admitted to ICU which was 67.8%, on day 7 was 22.3% vs 28.6%. These results are not statistically significant (log-rank p = 0.395) (Fig 3).

The probability of survival at 3 days of ICU stay was 65.3%, at 7 days 24.2%, and 0% on day 14. In patients with sepsis, the probability of survival on day 2 was 71.4% compared to those who were not admitted to the ICU, which was 62.5%; on day 6 it was 35.7% vs. 12.5%. This result is not statistically significant (log-rank p = 0.22), nor was statistical significance found with other complications such as ARDS with SAFI<150 (log-rank p = 0.19) (Fig 4).

### 3.4. Multivariate analysis

Poisson regression models have been created with bivariate associated variables, obtaining three initial models with epidemiological characteristics (hypertension, diabetes, cough, previous use of corticoids, previous use of anticoagulation, and respiratory frequency on

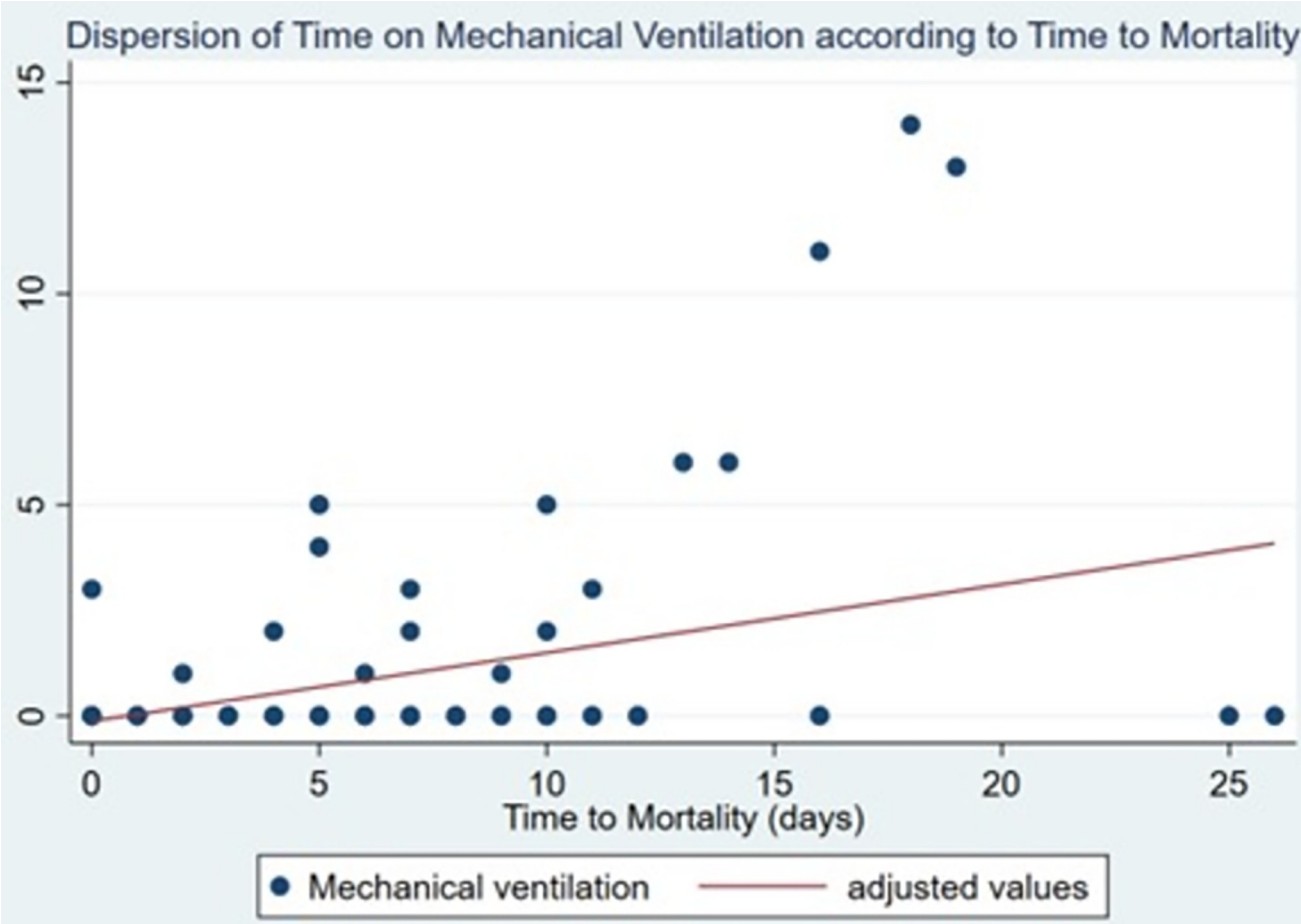

**Fig 1. Time on mechanical ventilation according to time to mortality.**

admission), clinical and laboratory characteristics (lymphocytes, CRP, DD, type of infiltrating, percentage of involvement), in addition to the therapeutic, complications and hospital times (corticotherapy, therapeutic anticoagulation, ICU admission criteria, ARDS due to SAFI, sepsis and time in ICU). Variables were eliminated from the models due to their high degree of collinearity and the models were designed. Subsequently, a single model was created with the verified variables. The numerical variables analyzed, and their respective categories were not taken into account.

Poisson regression showed that the probability of death in adult patients hospitalized for COVID-19 increased with comorbidities such as hypertension (PR:1.45; 95%CI: 1.01–2.09), with imaging findings of crazy paving (PR:1.51: 95%CI 1.1–2.09) and in patients with complications such as sepsis (PR:9.47; 95%CI: 3.72–24.07) (Table 4).

## 4. Discussion

The COVID-19 pandemic remains a public health emergency of global significance that threatens human health and welfare [19]. Therefore, in this retrospective cohort study, clinical and laboratory data from 300 confirmed cases of COVID-19 admitted to the Hospital Nacional Almanzor Aguinaga Asenjo (HNAAA) in Peru were analyzed.

Mortality in the study was 51.18 per 100 persons. In Peru, Valenzuela K et al. reported a mortality rate of 71.83 per 100 persons [20]. However, it differs from that reported in France

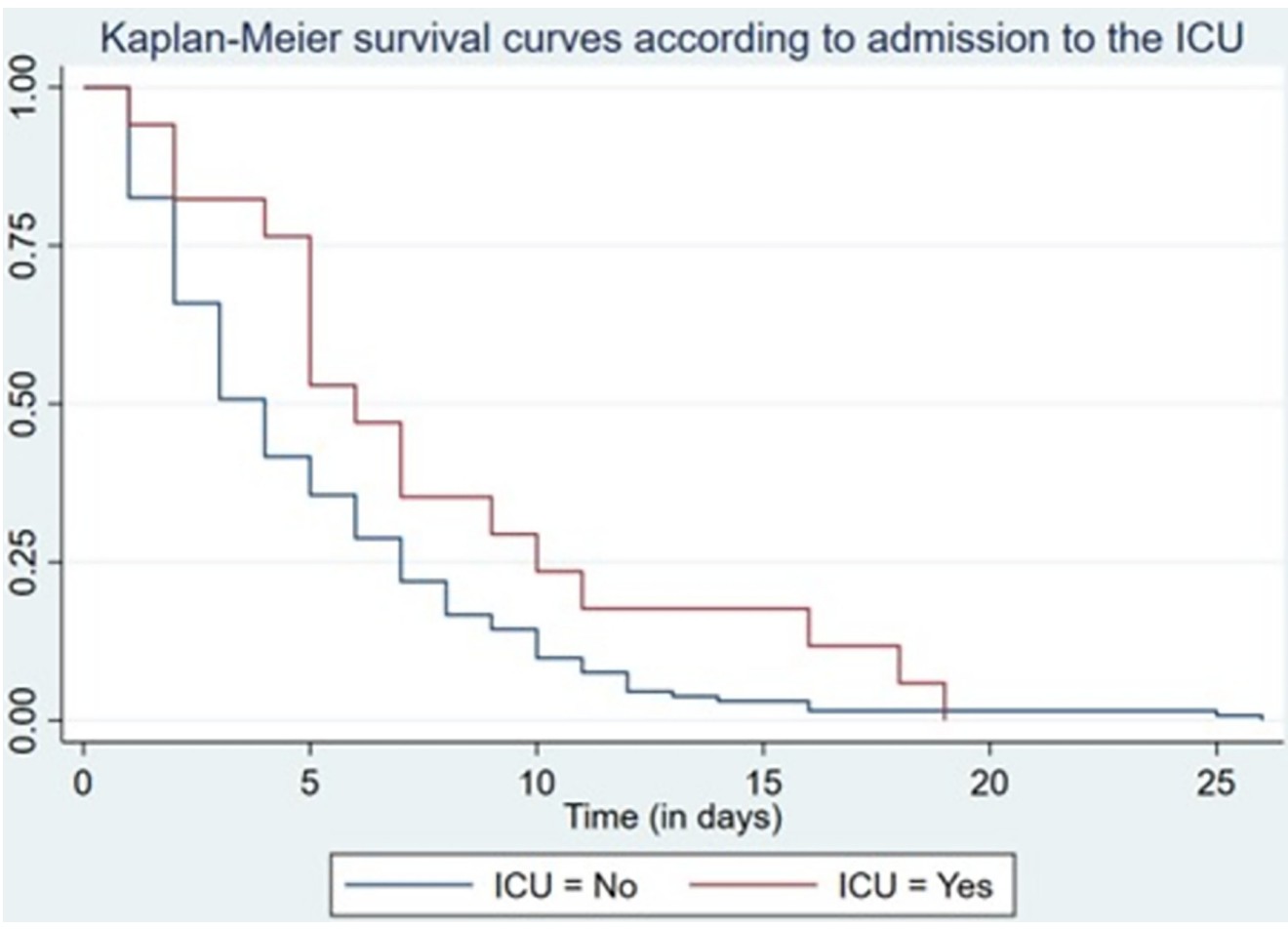

**Fig 2. Kaplan-Meier survival according to admission to the ICU.**

by Martinot M, et al, who reported lower mortality, reaching 19.1% [21]. This could be explained by demographic and clinical differences, comorbidities (hypertension), and specific treatments that were discovered based on evidence throughout the pandemic, as well as clear differences in the health systems in place before the health emergency [22, 23].

The present study demonstrated according to the regression model that the probability of mortality was higher in persons with a history of arterial hypertension, in those who had imaging findings such as consolidation or crazy paving, and who had complications such as sepsis. These models are comparable with other studies such as that of Filardo T et al. In New York, where mortality was associated with dementia (RR 2.11 95%CI 1.50–2.96), age 65 years or older (RR 1.97, 95%CI 1.31–2.95), obesity (RR 1.37, 95%CI 1.07–1.74) and male sex [24], Mendes A et al, in an older adult population in Switzerland with a model composed of the male sex (HR 4.00, 95% CI 2.08–7.71), crepitant (HR 2.42, 95% CI 1.15–6.06), and increased FiO2 use (HR 1.06, 95% CI 1.03–1.09) [25], Kamis F et al in Oman in their study COVID-19 mortality was associated with advanced age, heart disease (aOR: 1.84; 95% CI: 1.11–3.03), liver disease, elevated ferritin, ARDS, sepsis (aOR: 1.77; 95% CI: 1.12–2.80), and ICU admission (aOR: 2.22; 95% CI: 1.12–4.38) [26], or that of Huan J et al in China where associated factors such as C-reactive protein (HR, 2.063; 95% CI, 1.036–4.109), neutrophil count (HR, 2.015; 95% CI, 1.154–3.518), and interleukin 6 were found [27].

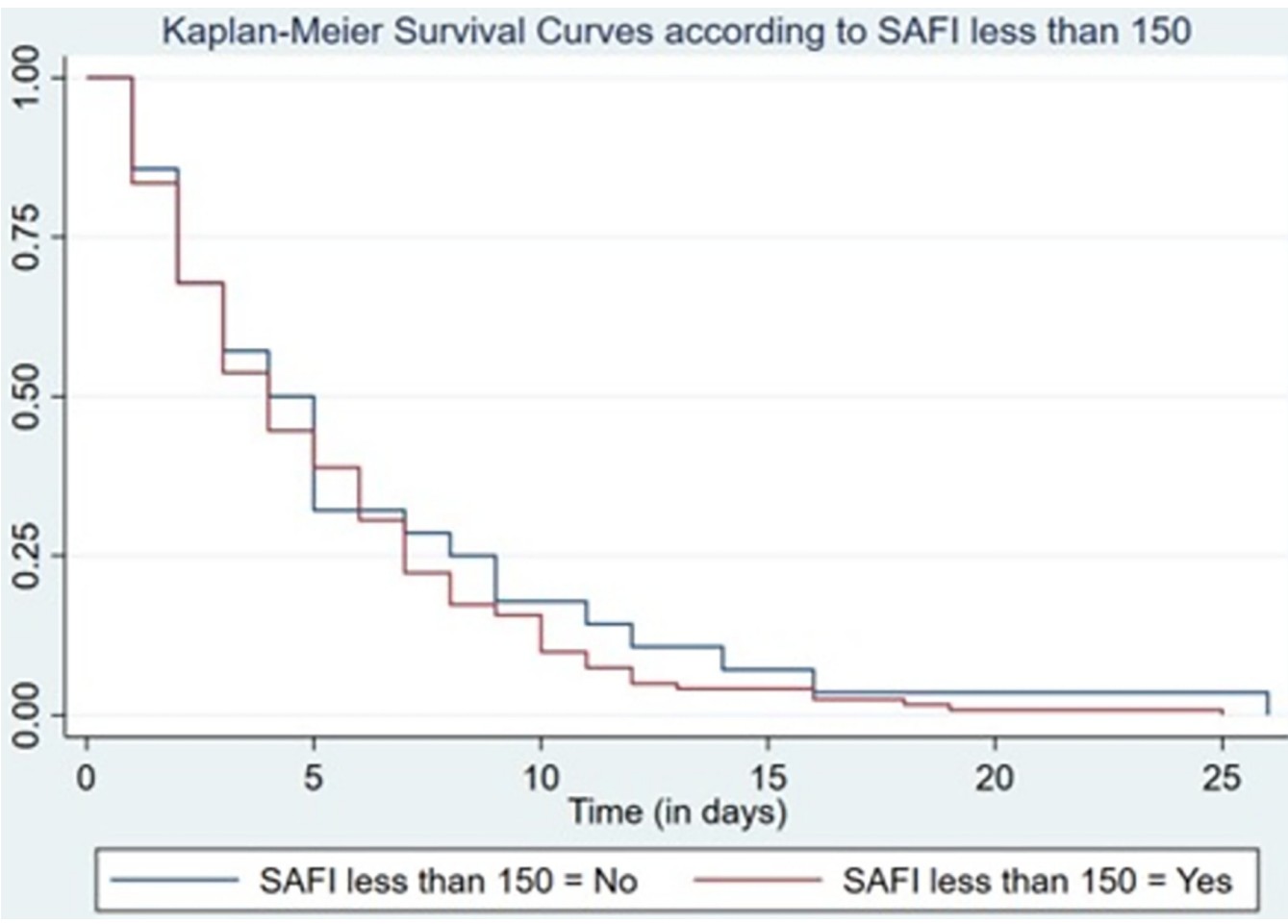

**Fig 3. Kaplan-Meier Survival according to SAFI less than 150.**

Hypertension is the leading cause of cardiovascular disease and death worldwide, increasing the risk of in-hospital deaths, ICU admissions, and the need for invasive ventilation in patients with COVID-19 [27, 28]. The results presented in our study show similar results, the mortality of patients with COVID-19 was higher in hypertensive individuals. However, the strength of association is lower than reported in other studies. Mubarik S, et al in China, presented similar results, where patients with hypertension had twice the risk of COVID-19 mortality, in addition, the median survival time of 3 to 5 days was shorter than in patients without hypertension [28].

Although there is no clear evidence regarding the association between hypertension and COVID-19 mortality, many mechanisms could explain this association. Patients with underlying cardiovascular disease are more likely to decline to an unstable hemodynamic state [29]. The inflammatory reaction caused by SARS-CoV-2 infection could lead to a hypercoagulant state and ruptures of atherosclerotic plaques, culminating in thrombotic events [30]. The priority of treatments administered in the ICU is focused on reducing the inflammatory response produced by SARS-CoV-2 and nosocomial infections, neglecting other conditions, which could lead to unfavorable clinical outcomes for patients with cardiovascular diseases [31].

The study showed that sepsis was a predictor of mortality. The study by Benes J. et al. in 2022 reported that sepsis and respiratory failure accounted for about 75% of deaths in patients

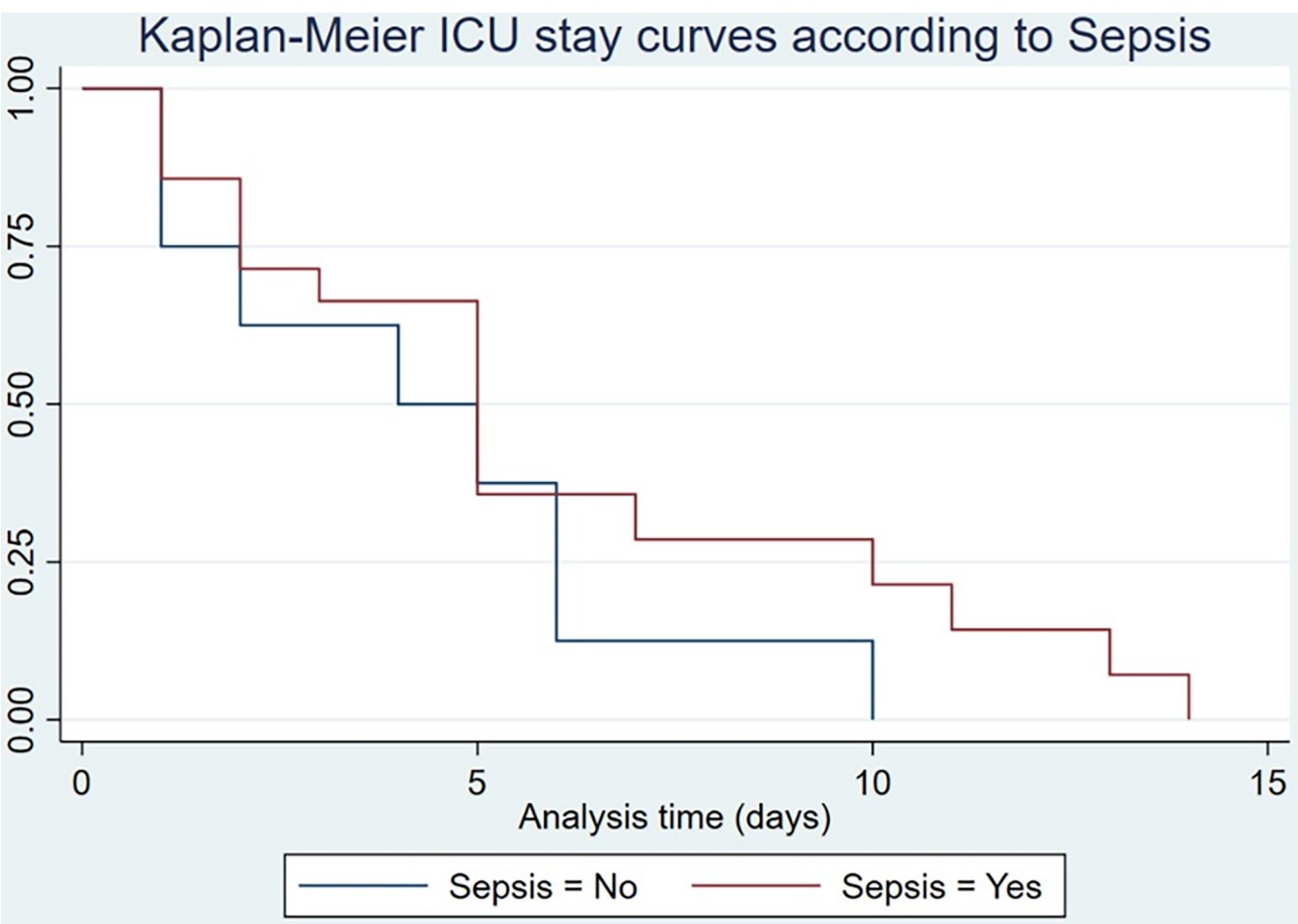

**Fig 4. ICU stay by Kaplan-Meier by Sepsis.**

with COVID-19; as a consequence, they increased ICU stays, necessitated the administration of vasopressor and inotropic drugs, and led to renal impairment [32, 33]. This could be explained by the fact that sepsis results in life-threatening organ dysfunction caused by a dysregulated host response to COVID-19 infection [34]. To identify septic patients at risk of in-hospital mortality, factors such as the need for invasive ventilation and renal replacement therapy, the presence of septic shock, and procalcitonin levels greater than 2 ng/ml should be considered [35, 36].

On the other hand, findings of specific infiltrates such as crazy paving were found to be associated with increased mortality. Currently, it has been found that radiographic infiltrate findings, especially bilateral ones, together with other factors such as age, comorbidities (hypertension, chronic respiratory disease, asthma, chronic heart disease, and chronic kidney disease), and biomarkers are associated with worse outcomes, including mortality and a prolonged hospital stay [37]. The imaging evolution of severe disease suggests that these findings would be very frequent in the bilateral distribution of lung disease.

It has been previously described that with advancing age, mortality for COVID-19 is increasing [5]. Our study was no exception, with a mean age of 63.99 in deceased patients. This coincides with previous studies carried out at the national level [18, 38] and internationally [24, 26].

**Table 4. Factors associated with mortality in patients hospitalized for COVID-19.**

| Variable | PR | P-value | 95% CI |
|---|---|---|---|
| **Arterial hypertension** | | | |
| No | Ref. | | |
| Si | 1.45 | 0,046 | 1.01–2.09 |
| **Age** | | | |
| 18–59 | Ref. | | |
| >60 | 1.46 | 0.082 | 0.95–2.25 |
| **Respiratory frequency** | | | |
| <30 | Ref. | | |
| >30 | 0.97 | 0.929 | 0.62–1.54 |
| **Type of infiltrate** | | | |
| Ground glass | Ref. | | |
| Cobblestone | 1.52 | 0.011 | 1.1–2.09 |
| Consolidation | 1.22 | <0.0001 | 2.70–5.53 |
| **Sepsis** | | | |
| No | Ref. | | |
| Yes | 9.47 | <0,0001 | 3.72–24.07 |

PR: potential ratio; CI: Confidence interval

This could be based on the fact that a high number of older adults develop a state of vulnerability due to mild chronic inflammation, which leads to frailty and adverse outcomes such as mortality [39]. This coupled with the accumulation of senescent cells, which secrete pro-inflammatory mediators [40], would further increase inflammation in COVID-19-infected patients, thereby increasing mortality, as hypothesized by Akbar AN. et al. [41].

Some laboratorial characteristics were found to be associated with COVID-19 mortality, among which D-dimer, LDH, and PCR were mainly associated. Similar results were reported by other studies conducted in China [42] and the United States [43]. In the Peruvian population, an association was found with CRP, LDH> 350 UI/L, ferritin> 750 ng/mL, and leukocyte. In addition to D-dimer ($> = 1$ μg/mL), severe lymphopenia, both in crude and adjusted analyses [44, 45].

LDH is an enzyme found inside cells in almost all organ systems [43], increased levels may predict severity and mortality from COVID-19 [46]. Similarly, increased D-dimer, a product of fibrin degradation and an indicator of coagulopathy, has been associated with an increased risk of mortality in patients with COVID-19 [16, 47]. PCR is an inflammatory marker and is elevated in patients with severe COVID-19 disease [48–50].

Our study reported that there is an association between the administration of anticoagulants therapeutically and increased deaths in COVID-19-infected patients. This is consistent with a recent meta-analysis, which reports that administration of anticoagulants in persons with COVID-19 was not associated with a lower risk of all-cause mortality and ICU admission compared with those who were not given anticoagulants [51].

The range of incidence of thrombosis among hospitalized patients with COVID-19 is wide, between 7.7–16.0% in hospitalized patients and 6.7–29.4% of those admitted to the ICU [52–54]. Early in the COVID-19 pandemic, international guidelines recommended the administration of prophylactic low-molecular-weight heparin (LMWH) to all hospitalized patients with COVID-19 to reduce mortality [55].

Despite the adequate use of antithrombotic prophylaxis, recent studies have shown that patients with COVID-19 have a higher incidence of venous thromboembolism (VTE) than

those with other infectious diseases [56–58]. Reporting that in the first 24 hours after hospital admission, the incidence of VTE in patients with COVID-19 reaches 50% [53].

Excess COVID-19 lethality may be related to hypercoagulability and microthrombi, leading to occlusion of pulmonary capillaries [59, 60], so that anticoagulation therapy initiated at advanced stages of COVID-19 may be insufficient [61].

This could be explained by the fact that the hypercoagulability of COVID-19 is a product of vascular inflammation [62], Therefore, the application of anticoagulants does not influence the progression of COVID-19, nor the mortality of infected patients [63]. Based on this, the best treatment option is to focus on thrombogenic inflammation or vasculopathy and not on secondary hypercoagulability [64].

The results should be interpreted with caution. The characteristics of the selection of participants do not allow us to extrapolate the data to the study population; however, addressing this issue gives us an impression of the reality of hospital mortality due to COVID-19 during the first wave of the pandemic in Peru, the Lambayeque region being one of the first regions to face the first wave of COVID-19. Other factors not considered in the study that could influence mortality (genetic factors, environmental factors, interactions between therapeutic options, unmeasured variables, spurious associations) should be taken into consideration. The data were obtained from digital medical records, so we may have errors in the measurement and missing data on variables. In addition, we do not have the necessary information to calculate risk functions such as the Hazard Ratio. Prospective studies covering other variables not considered are recommended, in addition to assessing the applicability of the model in different phases of the pandemic to recognize patients at high risk of mortality and improve their access to critical care units.

## 5. Conclusions

In this cohort, mortality in hospitalized patients with COVID-19 was 51.18 per 100 persons and was associated with a history of hypertension, type of infiltrating, and sepsis.

## Acknowledgments

Thanks are due to Alejandro Juárez-Ubillus and Pierina Perez-Espinoza for their contributions to the initial versions of the study.

## Author Contributions

**Conceptualization:** Edwin Aguirre-Milachay, Darwin A. León-Figueroa, Marisella Chumán-Sánchez.

**Data curation:** Edwin Aguirre-Milachay.

**Formal analysis:** Edwin Aguirre-Milachay, Luccio Romani.

**Investigation:** Edwin Aguirre-Milachay.

**Methodology:** Edwin Aguirre-Milachay.

**Resources:** Darwin A. León-Figueroa, Marisella Chumán-Sánchez.

**Software:** Darwin A. León-Figueroa, Luccio Romani.

**Supervision:** Fernando M. Runzer-Colmenares.

**Validation:** Fernando M. Runzer-Colmenares.

**Writing – original draft:** Darwin A. León-Figueroa, Marisella Chumán-Sánchez, Luccio Romani.

**Writing – review & editing:** Edwin Aguirre-Milachay, Darwin A. León-Figueroa, Marisella Chumán-Sánchez.

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
