## [Decision Letter · Decision Letter 0]

14 Nov 2022

PONE-D-22-28266Factors associated with mortality in patients hospitalized for COVID-19 admitted to a tertiary hospital in Lambayeque, PeruPLOS ONE

Dear Dr. Aguirre,

Thank you for submitting your manuscript to PLOS ONE. After careful consideration, we feel that it has merit but does not fully meet PLOS ONE’s publication criteria as it currently stands. Therefore, we invite you to submit a revised version of the manuscript that addresses the points raised during the review process.

We have had difficulty finding available reviewers for your work, as many considered the perceived impact of the manuscript to be very low, however, PLoS ONE does not consider this a cause for rejection.

We look forward to receiving your revised manuscript.

Kind regards,

Kovy Arteaga-Livias

Academic Editor

PLOS ONE

“No”

“No”

6. Please include your tables as part of your main manuscript and remove the individual files. Please note that supplementary tables (should remain/ be uploaded) as separate "supporting information" files

Reviewers' comments:

Reviewer's Responses to Questions

**Comments to the Author**

1. Is the manuscript technically sound, and do the data support the conclusions?

Reviewer #1: Yes

Reviewer #2: Yes

Reviewer #3: Partly

2. Has the statistical analysis been performed appropriately and rigorously? 

Reviewer #1: Yes

Reviewer #2: Yes

Reviewer #3: N/A

3. Have the authors made all data underlying the findings in their manuscript fully available?

Reviewer #1: Yes

Reviewer #2: Yes

Reviewer #3: No

4. Is the manuscript presented in an intelligible fashion and written in standard English?

Reviewer #1: No

Reviewer #2: Yes

Reviewer #3: Yes

5. Review Comments to the Author

Reviewer #1: I am submitting my comments in reference to the manuscript “Factors associated with mortality in patients hospitalized for COVID-19 admitted to a tertiary hospital in Lambayeque, Peru”. After going through the manuscript I found.

1.) The data presented is very much relevant to the scientific community.

2.) The manuscript has many typing mistakes and errors in sentence formation eg. Line 160 Abbreviation is ARDS but its expansion is Severe Respiratory Distress Syndrome.

3.) Many others are there which cannot be elaborated here.

It is advised to get the manuscript checked by an expert for English correction and resubmit.

Reviewer #2: I suggest that the title include that the study was carried out in the first wave in order to differentiate it from future studies that determine the characteristics found in the second, third and fourth waves, since characteristics such as morbidity and mortality could be different.

Reviewer #3: This is a retrospective cohort study and clinical records from a level 3 hospital are analyzed.

A great limitation that I can perceive is that they only add the variables of sepsis and hypertension (in a brief way) compared to studies that were previously published. In the same way, they do not add a sample size calculation or a multivariate analysis.

I will comment according to the line number in the pdf.

Line 45. mention the date for this information

Line 63. What new information would this study be providing?

Line 76. Didn't they sample? there are previous studies that they mentioned and they can base their sample size calculation on those. Even the aforementioned study carried out in a public hospital in the city of Tacna calculates statistical power for its realization.

Line 88. Why don't they mention the sepsis criteria they are using? since in the results they mention a complete paragraph to the resolution of said variable.

Line 100. Why not perform a multivariate analysis? the result of arterial hypertension is probably diminished when adjusting according to age, other comorbidity and need for ICU

Line 106. Is there a registration number? please include the resolution of the ethics committee of this hospital in complementary files

Line 135. So, the results of 46 other patients?

Line 215. considered stratifying by age in order to assess arterial hypertension?

Line 239. It would be ideal if they could provide information on the size of the association in the results they are comparing.

Line 243. It is a hasty statement according to their results, keep in mind that their prevalence ratio has a limit of 1.01 to 2.09 and has a confidence interval of more than one point.

Line 256. Why do they make this statement? if in their results they mentioned that there are no statistically significant differences.

Line 275. There are multiple studies that assess age as a negative outcome factor, not only national ones.

Line 284. and Peru? The study previously published in the Peruvian journal of experimental medicine and public health also mentions some of its population parameters.

Line 324. There are previously published studies that evaluate the variables of this study, including some prospective cohorts with a larger sample size, with multivariate analysis. Here some examples.

https://journals.plos.org/plosone/article?id=10.1371/journal.pone.0265089

https://pesquisa.bvsalud.org/global-literature-on-novel-coronavirus-2019-ncov/resource/zh/covidwho-1320619

https://researchonline.lshtm.ac.uk/id/eprint/4664503/

https://pubmed.ncbi.nlm.nih.gov/34292921/

6. PLOS authors have the option to publish the peer review history of their article (what does this mean?). If published, this will include your full peer review and any attached files.

Reviewer #1: No

Reviewer #2: **Yes: **Alberto Guevara Tirado

Reviewer #3: No

---

## [Author Response · Author response to Decision Letter 0]

12 Jan 2023

Dear reviewers, thank you for your comments and recommendations that have improved the quality of the article. In the following paragraphs, we will report on the lifting of comments and acceptance of suggestions.

FIRST REVIEWER

1. Reviewer comment: I am submitting my comments in reference to the manuscript “Factors associated with mortality in patients hospitalized for COVID-19 admitted to a tertiary hospital in Lambayeque, Peru”. After going through the manuscript I found.

Our response: “Thank you very much for your comments and recommendations, they have improved the quality of the article”

2. Reviewer comment: 1.) The data presented is very much relevant to the scientific community. The manuscript has many typing mistakes and errors in sentence formation eg. Line 160 Abbreviation is ARDS, but its expansion is Severe Respiratory Distress Syndrome.

Our response: “Thanks for your recommendation. The study was reviewed by an English language expert. Checking the grammar, and coherence of the sentences and terms used”

3. Reviewer comment: 3.) Many others are there which cannot be elaborated here.

Our response: “The article was again revised and updated, and the English version was improved. A comparison with high impact published studies was made”

4. Reviewer comment: It is advised to get the manuscript checked by an expert for English correction and resubmit.

Our response: “Carried out as recommended”

SECOND REVIEWER

1. Reviewer comment: I suggest that the title include that the study was carried out in the first wave in order to differentiate it from future studies that determine the characteristics found in the second, third and fourth waves, since characteristics such as morbidity and mortality could be different.

Our response: “Thank you very much for your comments and recommendations. The title has been corrected and the suggestion made has been added”

THIRD REVIEWER

This is a retrospective cohort study and clinical records from a level 3 hospital are analyzed.

A great limitation that I can perceive is that they only add the variables of sepsis and hypertension (in a brief way) compared to studies that were previously published. In the same way, they do not add a sample size calculation or a multivariate analysis.

I will comment according to the line number in the pdf.

1. Reviewer comment: Line 45. mention the date for this information.

Our response: “Added the date and updated the information: By December 1, 2022, the World Health Organization (WHO) has reported more than six million deaths due to COVID-19, in Peru, more than 217,370 deaths due to the disease have been reported by 2022, with an estimated case fatality rate of 5.14%”

2. Reviewer comment: Line 63. What new information would this study be providing?

Our response: “The importance and relevance of the study focused on recognizing the reality of hospital mortality due to COVID-19 during the first wave of the pandemic in Peru, with the Lambayeque region being one of the first to face the first wave of COVID-19”

3. Reviewer comment: Line 76. Didn't they sample? there are previous studies that they mentioned, and they can base their sample size calculation on those. Even the study carried out in a public hospital in the city of Tacna calculates statistical power for its realization.

Our response: “Thank you for your recommendation. The sample size was added and specified how it was determined and the study that was taken as a basis.”

4. Reviewer comment: Line 88. Why don't they mention the sepsis criteria they are using? since in the results they mention a complete paragraph to the resolution of said variable.

Our response: “Thank you for your comment. The sepsis criteria have been added in detail.”

5. Reviewer comment: Line 100. Why not perform a multivariate analysis? the result of arterial hypertension is probably diminished when adjusting according to age, other comorbidity and need for ICU

Our response: “Multivariate analysis of the Poisson regression type was performed because the sample size was not so large. Making a robust crude model and a final fitted model”

6. Reviewer comment: Line 106. Is there a registration number? please include the resolution of the ethics committee of this hospital in complementary files

Our response: “The ethics committee approval identifier number was added: The protocol of the present study was approved by the Research Ethics Committee of the Hospital Nacional Almanzor Aguinaga Asenjo, Lambayeque-Perú. The registration number is CIE-RPL:071-DIC-2021”

7. Reviewer comment: Line 135. So, the results of 46 other patients?

Our response: “The diagnosis of Covid-19 was mostly made with the rapid test in 188 patients, PCR-RT in 62, and antigenic test in 1 patient. The remaining patients were probable di-agnoses based on clinical and imaging findings”

8. Reviewer comment: Line 215. considered stratifying by age in order to assess arterial hypertension?

Our response: “Age was taken as numerical and categorical variables stratifying, aggregating in an initial model with comorbidities. However, further models were then analyzed based on other characteristics and finally a final model was analyzed taking into account all variables that could fit the model and in which collinearity did not harm them.”

9. Reviewer comment: Line 239. It would be ideal if they could provide information on the size of the association in the results they are comparing.

Our response: “Information on the size of the association was provided in the results that are comparing”

10. Reviewer comment: Line 243. It is a hasty statement according to their results, keep in mind that their prevalence ratio has a limit of 1.01 to 2.09 and has a confidence interval of more than one point.

Our response: “It was corrected according to your recommendation.”

11. Reviewer comment: Line 256. Why do they make this statement? if in their results they mentioned that there are no statistically significant differences.

Our response: “Corrected focus and wording: The aim is to explain that arterial hypertension framed within cardiovascular disease is a comorbidity whose man-agement and evaluation is left aside in the intensive care unit, but which could have implications in the evolution of the disease.”

12. Reviewer comment: Line 275. There are multiple studies that assess age as a negative outcome factor, not only national ones.

Our response: “National and international studies evaluating age as a negative outcome factor were added to the discussion.”

13. Reviewer comment: Line 284. and Peru? The study previously published in the Peruvian journal of experimental medicine and public health also mentions some of its population parameters.

Our response: “The study was reviewed and added to the discussion because of the impact and information it reported. ”

14. Reviewer comment: Line 324. There are previously published studies that evaluate the variables of this study, including some prospective cohorts with a larger sample size, with multivariate analysis. Here some examples.

( https://journals.plos.org/plosone/article?id=10.1371/journal.pone.0265089 ;

https://pesquisa.bvsalud.org/global-literature-on-novel-coronavirus-2019-ncov/resource/zh/covidwho-1320619

https://researchonline.lshtm.ac.uk/id/eprint/4664503/ ; 

https://pubmed.ncbi.nlm.nih.gov/34292921/ )

Our response: “The recommended studies were reviewed and added to the discussion.”

---

## [Decision Letter · Decision Letter 1]

16 Apr 2023

Factors associated with mortality in patients hospitalized for COVID-19 admitted to a tertiary hospital in Lambayeque, Peru, during the first wave of the pandemic.

PONE-D-22-28266R1

Dear Dr. Aguirre,

We’re pleased to inform you that your manuscript has been judged scientifically suitable for publication and will be formally accepted for publication once it meets all outstanding technical requirements.

Kind regards,

Kovy Arteaga-Livias

Academic Editor

PLOS ONE

Additional Editor Comments (optional):

Reviewers' comments:

Reviewer's Responses to Questions

**Comments to the Author**

1. If the authors have adequately addressed your comments raised in a previous round of review and you feel that this manuscript is now acceptable for publication, you may indicate that here to bypass the “Comments to the Author” section, enter your conflict of interest statement in the “Confidential to Editor” section, and submit your "Accept" recommendation.

Reviewer #2: All comments have been addressed

2. Is the manuscript technically sound, and do the data support the conclusions?

Reviewer #2: Yes

3. Has the statistical analysis been performed appropriately and rigorously? 

Reviewer #2: Yes

4. Have the authors made all data underlying the findings in their manuscript fully available?

Reviewer #2: Yes

5. Is the manuscript presented in an intelligible fashion and written in standard English?

Reviewer #2: Yes

6. Review Comments to the Author

Reviewer #2: El autor ha subsanado las sugerencias planteadas por el revisor, se sugiere la publicacion del articulo

7. PLOS authors have the option to publish the peer review history of their article (what does this mean?). If published, this will include your full peer review and any attached files.

Reviewer #2: **Yes: **Alberto Guevara Tirado

---

## [Editor Report · Acceptance letter]

4 May 2023

PONE-D-22-28266R1 

Factors associated with mortality in patients hospitalized for COVID-19 admitted to a tertiary hospital in Lambayeque, Peru, during the first wave of the pandemic. 

Dear Dr. Runzer-Colmenares:

I'm pleased to inform you that your manuscript has been deemed suitable for publication in PLOS ONE. Congratulations! Your manuscript is now with our production department. 

Kind regards, 

on behalf of

Dr. Kovy Arteaga-Livias 

Academic Editor

PLOS ONE